# Bootstrap Prompt Learning with Feature Adaptation for Vision-Language Efficient Tuning

## Abstract

Prompt learning is widely adopted for fine-tuning vision-language foundation models such as CLIP and offers strong generalization ability by inserting learnable embeddings in the input space for pre-adjustment. However, existing methods usually suffer from limited fitting capacity and heavily rely on biased exclusive cross entropy loss that compromises the generalization to unseen classes. To address these problems, in this paper, we propose the first framework named ada**P**ter bootstr**A**pped prompt contrastive **T**uning (PAT) to integrate the superior fitting capacity of post-adjustment via adapters into prompt learning. Specifically, we bootstrap prompt learning with adapters and achieves pre-post alignment to achieve a more effective trade-off between fitting capability and generalization ability. Furthermore, we propose a tolerance regularization that equally pushes away all negative samples and improves generalization by introducing additional categories of unlabeled data to avoid overfitting. To our best knowledge, this is the first successful attempt to simultaneously exploit the advantages of prompt learning and adapter tuning. Extensive evaluations demonstrate that PAT achieves state-of-the-art performance in various recognition tasks on three prevailing benchmarks.

## 1 Introduction

Vision-language models (VLMs) pretrained on large-scale image-text pairs have demonstrated strong representational capabilities Alayrac et al. (2022); Radford et al. (2021); Jia et al. (2021).Nevertheless, fine-tuning these models for downstream tasks demands substantial computational resources. Recently, parameter-efficient fine-tuning (PEFT) Han et al. (2024) has been emerging as a promising alternative to address these challenges. Compared to full fine-tuning, PEFT achieves competitive performance by tuning a minimal number of trainable parameters, therefore widely adopted as alternatives to full fine-tuning. Currently, the predominant approaches to PEFT could be generally categorized into three types, *i.e.*, prompt learning Jia et al. (2022), adapters Chen et al. (2022), and reparameterization Hu et al. (2022). Reparameterization based methods such as LoRA Hu et al. (2022) achieve low-rank decomposition of weight matrices for PEFT. Prompt learning introduces trainable embeddings into the input space to guide the adaptation of pretrained models to downstream tasks, while adapter-based methods insert trainable parameters alongside the original weight matrices.

Prompt learning is prevailing in the context of adapting VLMs. CoOp Zhou et al. (2022b) converts the text encoder into a classifier by combining classification labels with a classification template and introducing trainable text embeddings. Although prompt learning exhibits strong generalization ability, they remain limited in fitting ability. By contrast, adapter-based methods are less explored in the context of adapting VLM due to limitation in generalization, but they enjoy strong fitting ability. To this end, we employ Prompt learning as a form of pre-alignment, and further enhance its effectiveness through Adapter-based post-alignment, striving to simultaneously achieve strong fitting capacity and robust generalization ability.

In Figure 1, we reveal that adapters emphasize fitting ability and prompt learning focuses more on generalization, and validate their discrepancy in fitting and generalization ability to adapt VLMs in base-to-new generalization. For adapters, we adopt commonly used AdapterFormer Chen et al.

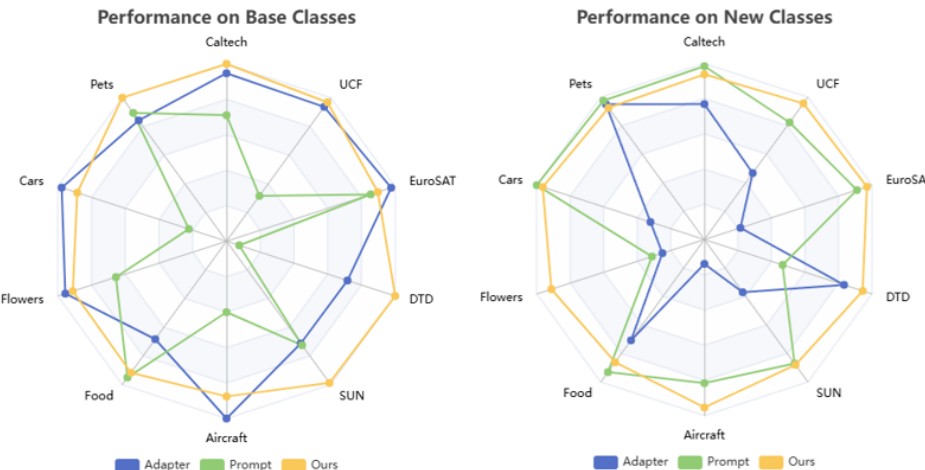

Figure 1: Evaluation on Base-to-New Experiments. The results on the base classes reflect the fitting capability of each method, while the results on the new classes indicate their generalization ability. Adapter-based methods possess stronger fitting capability but weaker generalization, while Prompt-based methods show the opposite tendency. In contrast, our method successfully achieves both strong fitting and generalization abilities.

(2022) for the text branch and Convpass Jie et al. (2024) for the vision branch of CLIP, and use the representative MaPLe approach as a representative for prompt learning. Adapter-based methods focus on fitting and is superior on base classes, while prompt learning favors generalization and outperforms adapters on new classes. Unfortunately, despite of the potential of integrating adapters and prompt learning, there still lacks an efficient paradigm to unify them for jointly enhanced fitting and generalization. To further enhance the model's generalization, we revisit the training paradigm of VLMs during transfer to downstream tasks and find that generalization to unseen categories is undermined in existing methods, since they rely heavily on exclusive cross entropy loss to select the class with the highest similarity between fine-tuned visual and textual representations as final prediction. Table 1 shows that the forced-choice constraint causes incorrect bias toward the categories given in the few-shot tuning and results in a loss of information about unseen categories.

In this paper, we propose a novel vision language efficient tuning framework named adaPter boot-strAp prompt contrastive Tuning (PAT) that for the first time simultaneously exploits the advantages of prompt learning and adapter tuning. PAT incorporates two novel modules, *i.e.*, pre-post alignment to match and integrate the prompt learning based pre-adjustment and adapter based post-adjustment and tolerance regularization to mitigate the bias caused by exclusive cross-entropy loss. Our contributions are summarized as below.

- We develop the adaPter bootstrAp prompt contrastive Tuning (PAT) framework to simultaneously improve the fitting capability and generalization performance of prompt learning in downstream tasks.

- We bootstrap the pre-adjustment with prompt learning by integrating post-adjustment with adapters and introduce a pre-post alignment module to integrate the fitting capability of Adapters with the generalization ability of Prompts.

- We propose a tolerance regularization, which equally pushes away all negative samples and improves generalization by introducing additional categories of unlabeled data to prevent the model from over-fitting on the training categories.

Extensive evaluations have been made to demonstrate the fitting and generalization abilities of PAT, including base-to-new generalization, few-shot learning, cross-dataset generation across more than ten datasets and comprehensive ablation studies. PAT is shown to achieve state-of-the-art performance compared with most recent vision language efficient tuning methods. Specifically, PAT achieves an average 0.9% accuracy gain in Base-to-New over existing state-of-the-arts (*i.e.* 80.4% vs. 79.5%), and achieves an average 1.5% improvement in the few-shot learning (*i.e.*, 78.2% vs. 76.7%).

## 2 RELATED WORK

### 2.1 VISION LANGUAGE MODELS

Vision-language models (VLMs) learn multi-modal representations by pretraining on large-scale image-text datasets, such as CLIPRadford et al. (2021) and ALIGNJia et al. (2021), with 400 million and 1 billion pairs, respectively. Using contrastive loss, these models align paired features while distinguishing unpaired ones, enabling strong open-vocabulary generalization. Recent advancements enhance their descriptive and discriminative capabilities through stronger encodersLi et al. (2023); Vaswani et al. (2017), deeper modality fusion, larger datasets, and techniques like Masked Language Modeling (MLM) and image maskingKim et al. (2021); Lu et al. (2019). CLIP, a key framework with exceptional generalization, has inspired numerous CoOp-based prompt tuning approaches. In this work, we propose a novel prompt learning framework to further adapt pretrained CLIP for generalization and few-shot learning.

### 2.2 PEFT FOR VISION LANGUAGE MODELS

Prompt learning, as a parameter-efficient fine-tuning method, aims to transfer pretrained models to downstream tasks while keeping most parameters frozen. Classical prompt learning methods achieve this by adding a small number of trainable embeddings into the input space of pretrained models without altering the pretrained weights, thereby guiding the model's outputs to adapt to downstream tasks. Due to its efficiency in terms of trainable parameters, developing more powerful prompt learning methods for adapting multimodal pretrained models like CLIP to visual or vision-text downstream tasks has garnered significant interest from both academia and industry. For example, Context Optimization (CoOp) Zhou et al. (2022b) replaces handcrafted prompts with learnable embeddings in the input space of CLIP's text encoder to enable few-shot adaptation. Recently, Textual-based Class-aware Prompt (TCP) Yao et al. (2024) proposed another paradigm, focusing on class-aware prompt tuning and try to combine adapter and prompt learning. To mitigate potential knowledge forgetting during fine-tuning, Knowledge-Guided Context Optimization (KgCoOp) Yao et al. (2023) applies L2 norm constraints to the text encoder, thus enhancing generalization.

Unlike these methods, we observe that while both are parameter-efficient fine-tuning approaches, adapter-based methods differ from prompt learning in their focus. Instead of modifying the input space of pretrained models as prompt learning does, adapter-based methods insert a small number of trainable parameters alongside the pretrained modules. This indicates that these two approaches exhibit different tendencies in generalization and fitting when learning knowledge. In this paper, we propose a novel approach that leverages prompt learning as a pre-adjustment, followed by a post-adjustment using adapter methods. By aligning the representations learned from both approaches, we demonstrate that the knowledge acquired through adapter methods can be utilized to further bootstrap the effectiveness of prompt learning.

## 3 METHODOLOGY

### 3.1 REVISITING VISION-LANGUAGE MODEL

We consider the pre-trained vision-language model CLIP that comprises a text encoder $g$ and a vision encoder $f$ with respective pre-trained parameters $\theta_g$ and $\theta_f$. We denote $\theta_{CLIP} = \{\theta_g, \theta_f\}$ as the collection these parameters.

**Vision Encoder:** An input image $X \in \mathbb{R}^{C \times H \times W}$ is first divided into $M$ patches that are projected into $M$ patch tokens $t_1, \cdots, t_M$. The input $\hat{X} = \{t_{cls}, t_1, \cdots, t_M\}$ to the vision encoder $f$ is then formed by appending a learnable class token $t_{cls}$ to the $M$ patch tokens. Latent visual feature representation $\hat{f} = f(\hat{X}, \theta_f) \in \mathbb{R}^d$ is extracted from $\hat{X}$ with multiple transformer blocks.

**Text Encoder:** The class label $y$ corresponding to the image is wrapped within a text template (*e.g.*, 'a photo of a class label') to form $\hat{Y} = \{t_{SOS}, t'_1, \cdots, t'_L, c_k, t_{EOS}\}$, where $\{t'_l\}_{l=1}^L$ and $c_k$ are word embeddings for the text template and class label of the $k_{th}$ class, respectively, and $t_{SOS}$ and $t_{EOS}$ are learnable start and end token embeddings. The text encoder $g$ encodes $\hat{Y}$ via multiple transformer blocks to obtain the latent textual feature $\hat{g} = g(\hat{Y}, \theta_g) \in \mathbb{R}^d$.

Table 1: In the Base-to-New experiment, performance comparison between classification experiments using New classes labels and All classes labels on the New classes dataset. Using All classes labels results in a significant performance loss.

| Datasets | Labels | CoOp | CoCoOp | MaPLe | PromptSRC |
|---|---|---|---|---|---|
| SUN397 | New | 68.3 | 76.9 | 78.7 | 79.0 |
|  | All | 57.9 | 67.4 | 69.0 | 68.6 |
| EuroSAT | New | 53.0 | 60.0 | 73.2 | 68.4 |
|  | All | 41.7 | 49.4 | 46.3 | 54.6 |
| UCF101 | New | 67.4 | 73.5 | 78.7 | 78.3 |
|  | All | 52.3 | 65.6 | 71.3 | 71.6 |

**Zero-shot Classification for Vision-Language Model:** For zero-shot classification, textual prompts are crafted with the text template and class labels $y \in \{1, \cdots, C\}$ for $C$ classes. The prediction $\hat{y}$ given the image feature $\hat{f}$ is calculated by cosine similarity with a temperature parameter $\tau$.

$$p(\hat{y}|\hat{f}) = \frac{\exp(sim(\hat{f}, \hat{g}_{\hat{y}})/\tau)}{\sum_{C}^{i=1} \exp(sim(\hat{f}, \hat{g}_i)/\tau)}. \tag{1}$$

**Limitations of Different Tuning Methods:** Prompt learning inserts trainable embeddings into the model's input space without modifying its internal parameters, which can lead to instability during training. Furthermore, since these embeddings merely guide the model's output, their effectiveness in downstream tasks is highly dependent on the pretrained model's inherent capabilities. Consequently, prompt learning performs poorly in scenarios where there is a significant distribution shift between the pretraining data and downstream tasks or when handling complex tasks. In contrast, adapter-based methods introduce trainable modules alongside the model's parameter matrices, enabling stronger representational capacity. They are more robust to distribution shifts and complex datasets; however, this learning tendency may result in a loss of the model's generalization ability. Therefore, an important research question is how to efficiently integrate adapter-based methods with prompt learning to leverage their respective advantages.

## 3.2 PROPOSED METHOD

Existing PEFT methods for vision-language models such as TCP Yao et al. (2024) and DePT Zhang et al. (2024), are primarily Prompt-based, thereby exhibiting strong generalization capabilities. However, as illustrated in Figure 1, prompt-based methods remain limited in fitting capacity, whereas Adapter-based methods demonstrate superior fitting ability. Accordingly, this section introduces PAT, which integrates Adapter and Prompt as pre-adjustment and post-adjustment, and propose a pre-post-alignment to enhance the coordination between the two method and promote more effective joint optimization. To futher reinforce the model's discriminative and generalization capability, we combined PAT with tolerance regulatization, ultimately achieving more robust classification performance. Figure 2 depicts the overall framework architecture. We use prompt learning as a pre-adjustment to fine-tune the pre-trained VLM, followed by adapter tuning as a post-adjustment, as formulated below.

$$\hat{f}_{\alpha} = f(\hat{X}, \{\theta_f, \alpha_f\}), \ \hat{g}_{\alpha} = g(\hat{Y}, \{\theta_g, \alpha_g\}), \tag{2}$$

$$\hat{f}_{\beta} = f(\{\beta_f, \hat{X}\}, \theta_f), \ \hat{g}_{\beta} = g(\{\beta_g, \hat{Y}\}, \theta_g), \tag{3}$$

$$\hat{f} = \hat{f}_{\alpha} + \hat{f}_{\beta}, \ \hat{g} = \hat{g}_{\alpha} + \hat{g}_{\beta}, \tag{4}$$

where $\alpha_g$ and $\alpha_f$ denote the learnable parameters of adapter inserted alongside the model for the text branch and the visual branch, respectively. $\beta_g$ and $\beta_f$ represent the prompt learnable parameters inserted into the input embeddings for the text branch and the visual branch, respectively. $\hat{f}_{\alpha}$ and $\hat{g}_{\alpha}$ represent the fine-tuned representations obtained after applying prompt learning (pre-adjustment) of text branch and visual branch. Similarly, $\hat{f}_{\beta}$ and $\hat{g}_{\beta}$ indicate the fine-tuned representations obtained after applying adapter tuning (post-adjustment). the final fine-tuned representations $\hat{f}$ and $\hat{g}$ are obtained by integrating both approaches. Figure 3 illustrates the tolerance regularization mechanism.

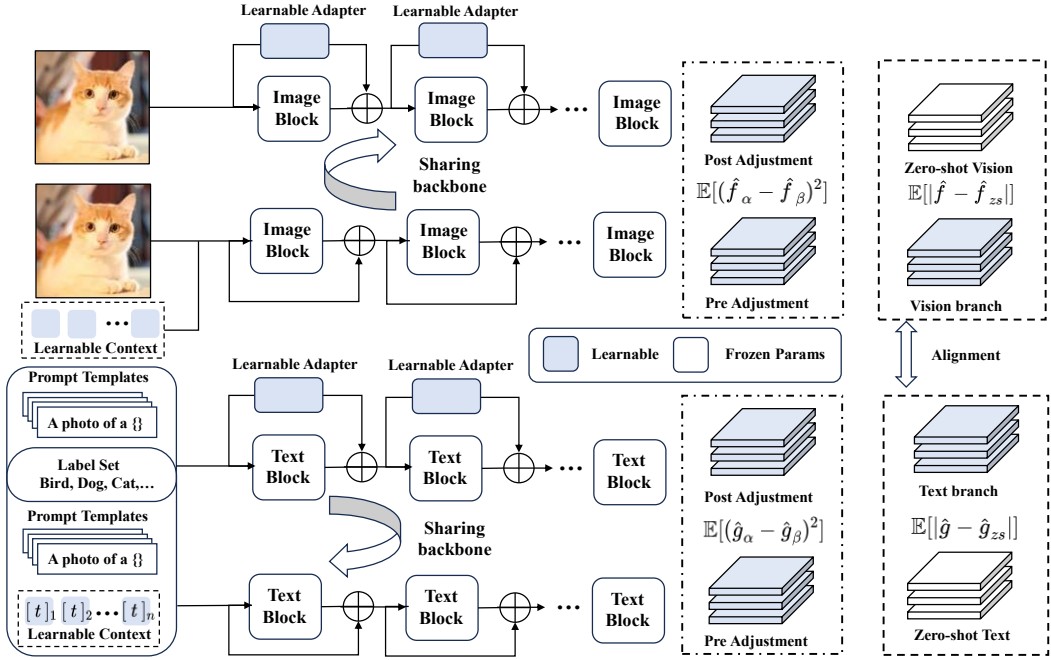

Figure 2: Overall Structure of PAT. In this figure, the blue blocks represent trainable parameters. As shown in the overall pipeline of PAT, trainable parameters are first inserted into the input space of both the image and text branches as pre-adjustment, followed by the insertion of adapters as post-adjustment. Then, MSE is applied to align this two features. Subsequently, the two representations are integrated through equal-weighted summation and the features after pre- and post-adjustments, together with the features obtained from Zeroshot CLIP are further constrained using MAE and KL divergence.

### 3.2.1 PRE-POST ALIGNMENT

To compensate for the limited fitting capacity of Prompt learning, a feasible approach is to constrain the features obtained by adapter and prompt, thereby promoting the optimization of Prompts through the updates of Adapters. Considering that $x$ is the input image, $z$ is the latent feature, $\alpha$ and $\beta$ is the parameterized adapter and prompt respectively. Then we have

$$p_\alpha(y|x) = \int p_\alpha(y|z)p_\alpha(z|x)\mathrm{d}z = \mathbb{E}_z[p_\alpha(y|z)], \tag{5}$$

$$p_\beta(y|x) = \int p_\beta(y|z)p_\beta(z|x)\mathrm{d}z = \mathbb{E}_z[p_\beta(y|z)]. \tag{6}$$

In equation 5 and equation 6, we assume that $p_\alpha(z|x)$ and $p_\beta(z|x)$ obey the Gaussian distributions $\mathcal{N}(z; \mu_\alpha(x), \sigma_\alpha^2 I)$ and $\mathcal{N}(z; \mu_\beta(x), \sigma_\beta^2 I)$. $p_\alpha(y|z)$ and $p_\beta(y|z)$ are the determinant function, *i.e.*, the linear projection layer

$$p_\alpha(y|z) = \delta(y - W_\alpha z), \quad p_\beta(y|z) = \delta(y - W_\beta z), \tag{7}$$

where $\delta(\cdot)$ is the Dirac delta function and $W_\alpha$ and $W_\beta$ are the weight matrices. The expectation can be simplified as

$$p_\alpha(y|x) = p_\alpha(z = \mu_\alpha(x)|x), \ p_\beta(y|x) = p_\beta(z = \mu_\beta(x)|x). \tag{8}$$

We aim to minimize the KL divergence between the prediction distribution as

$$\mathcal{D}_{KL}(p_\alpha(y|x)\|p_\beta(y|x)) = \mathbb{E}_{y \sim p_\alpha(y|x)}\left[\log \frac{p_\alpha(y|x)}{p_\beta(y|x)}\right]. \tag{9}$$

Then we bring Eq. equation 8 into the above objective. Considering that $p_\alpha(z|x)$ and $p_\beta(z|x)$ obey Gaussian distribution, then the KL divergence has the analytical form:

$$\mathcal{D}_{KL}(p_\alpha\|p_\beta) = \frac{1}{2}\left[\log \frac{\sigma_\beta^2}{\sigma_\alpha^2} + \frac{\sigma_\alpha^2 + (\mu_\alpha - \mu_\beta)^2}{\sigma_\beta^2} - 1\right]. \tag{10}$$

For simplification, we assume the adapter $\alpha$ and prompt $\beta$ have the same variance $\sigma_\alpha^2 = \sigma_\beta^2 = \sigma^2$ in equation 10. Thus, we obtain that

$$\mathcal{D}_{KL}(p_\alpha \| p_\beta) = \frac{1}{2\sigma^2} \|\mu_\alpha - \mu_\beta\|^2. \tag{11}$$

Therefore, the pre-post alignment loss for each modal branch model is formalized using MSE.

$$\mathcal{L}_{pre-post} = \mathbb{E}[(\hat{f}_\alpha - \hat{f}_\beta)^2] + \mathbb{E}[(\hat{g}_\alpha - \hat{g}_\beta)^2] \tag{12}$$

### 3.2.2 TOLERANCE REGULARIZATION

For each sample $x_i$, the corresponding visual feature is $f_i$, and for all the textual description, the textual representation is $\{t_k\}_{k=1}^K$, where $K$ is the number of categories. We then calculate the logits after softmax as

$$\hat{y}_i^{(k)} = \frac{\exp\left(cf_i t_k + b\right)}{\sum_{j=1}^K \exp\left(cf_i t_j + b\right)}. \tag{13}$$

where $c$ is the constant and $b$ is the bias. For one-hot label $y_i^{(k)} \in \{0, 1\}$, the cross-entropy loss is defined as

$$H(\hat{y}, y) = -\frac{1}{|B|} \sum_{i=1}^{|B|} \sum_{k=1}^K y_i^{(k)} \log \hat{y}_i^{(k)}. \tag{14}$$

In conventional vision language efficient tuning, the predicted textual description is forced to match one of the given label. This over-fits the training categories and hampers generalization to unseen classes. To avoid over-fitting on training datasets, we propose to use binary contrastive loss, *i.e.*, the sigmoid loss, as the regularization in the objective function.

$$\mathcal{L}_{tol}(f_i, g_j) = -\frac{1}{|\mathcal{B}|} \sum_{i=1}^{|\mathcal{B}|} \sum_{j=1}^{|\mathcal{B}|} \log \frac{1}{1 + \exp(z_{ij}(-tf_i \cdot g_j + b))}, \tag{15}$$

where $z_{ij}$ returns 1 when the $i$-th visual representation $f_i$ matches the $j$-th textual representation $g_j$ and $-1$ otherwise. Different from Zhai et al. (2023), we fix the parameters $b$ to $-2$ and $t$ to $-\log 2$, since the model possesses strong representation capability during fine-tuning.

**Theorem 1.** *The sigmoid loss function degenerates to the class-irrelevant binary cross entropy loss function, when considering only positive samples.*

*Proof.* Please refer to the appendix. □

Proposition 1 demonstrates that the proposed tolerance regularization yields a non-differentiable binary cross-entropy (BCE) loss. Furthermore, when incorporating images from unrelated categories without label information and randomly sampled mismatched text, the non-differentiable constraint effectively prevents the model from incorrectly assigning these samples to inappropriate categories. By avoiding the enforcement of erroneous category selection, the regularization enhances the model's robustness and generalization capability

As shown in Figure 3, the training data is divided into images within the current category space paired with their corresponding labels and noise images outside the current category space paired with randomly sampled labels during fine-tuning. The tolerance regularization processes each image-text pair independently. For images within the category space, it brings their representation closer to the corresponding textual representation. For noise images, the similarity with the representation of all existing category texts will be pushed farther. During this process, we progressively penalize the distance between unknown samples and known text representation, thereby enhancing their generalization to unseen categories. Subsequent experimental results demonstrate that the application of tolerance regularization significantly improves the generalization capability of prompt learning.

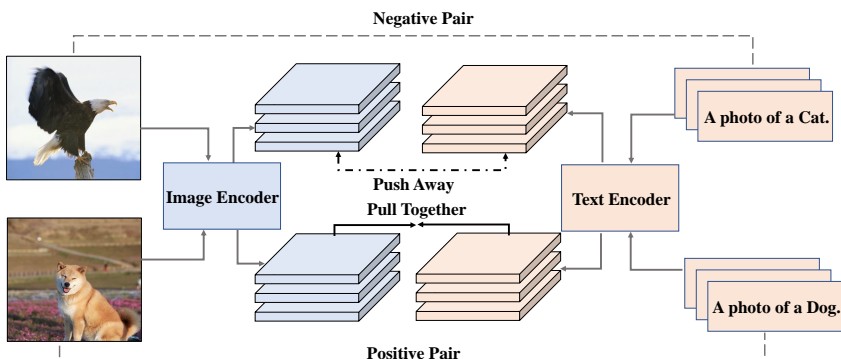

Figure 3: Schematic diagram of contrastive learning based on the tolerance regularization loss. For each image-text pair, if the pair is positive, the resulting visual representation and textual representation are pull together; otherwise, the two representations are pushed away apart.

### 3.2.3 OBJECTIVE FUNCTION

The final objective function of PAT is

$$\mathcal{L} = -0.01\mathcal{L}_{tol}(\hat{f}_i, \hat{g}_j^{(zs)}) + \mathcal{L}_{pre-post} + \mathcal{L}_{ce} + \mathcal{L}_{pre-tune}, \tag{16}$$

where $\hat{g}_j^{(zs)}$ represents the text representation obtained through zero-shot CLIP. In addition to the proposed pre-post align and pos-neg align, the objective function includes a cross-entropy loss $\mathcal{L}_{ce}$ for guiding classification and a widely adopted alignment loss $\mathcal{L}_{pre-tune}$ for prompt learning Khattak et al. (2023b); Yao et al. (2023; 2024) to constrain the pretrained and fine-tuned model.

## 4 EXPERIMENTS

### 4.1 BENCHMARK SETTINGS

**Datasets:** Following Khattak et al. (2023b) and Yao et al. (2024), we conduct Base-to-New generalization, few-shot learning, and cross-dataset generalization for a wide range of recognition tasks on 11 datasets, including ImageNet Deng et al. (2009) and Caltech101 Fei-Fei et al. (2004) for generic objects, OxfordPets Parkhi et al. (2012), StanfordCars Krause et al. (2013), Flowers102 Nilsback & Zisserman (2008), Food101 Bossard et al. (2014), and FGVC-Aircraft Maji et al. (2013) for fine-grained classification, SUN397 Xiao et al. (2010) for scene recoginition, UCF101 Soomro et al. (2012) for action recognition, DTD Cimpoi et al. (2014) for texture classification, and EuroSAT Helber et al. (2019) containing satellite images.

**Implementation Details:** We adopt the pretrained CLIP with the ViT-B/16 backbone. All the experiments are repeated for 3 times to report the average results. Under all the settings, the prompts are randomly initialized and trained for 20 epochs, and the length is set to 4. We adopt Adapter-Former Chen et al. (2022) for post-adjustment of the text encoder and Convpass Jie et al. (2024) for the vision encoder. For both modalities, the adapters are applied in the multi-head attention layer and the linear layer with a scaling factor of 0.1 and a hidden dimension of 16. For cross-dataset evaluation, we train the source model on all classes of ImageNet with 16 shots settings using SGD optimizer with the learning rate of $3.5 \times 10^{-3}$ and the batch size of 4. For feature ensembling, we add both feature from pre-adjustment and post-adjustment with equal weight. All the experiments are performed on NVIDIA RTX 2080Ti GPU except that evaluations on ImageNet are performed on RTX 4090 and NVIDIA A100 GPUs.

**Baselines:** We adopt most recent state-of-the-art methods without using the large language model as baselines, including CoOp Zhou et al. (2022b), CoCoOp Zhou et al. (2022a), ProGrad Zhu et al. (2023), KgCoOp Yao et al. (2023), PromptSRC Khattak et al. (2023b), MaPLe Khattak et al. (2023a), LFA Ouali et al. (2023), DePT Zhang et al. (2024), PLOT Chen et al. (2023), TaskRes Yu et al. (2023), RPO Lee et al. (2023), DAPTCho et al. (2023), VPT Jia et al. (2022), TIP-Adapter-F Zhang et al. (2022), TCP Yao et al. (2024) and DCP Li et al. (2025).

Table 2: Performance comparison across different methods on Base-to-New Benchmark. PAT achieved state-of-the-art performance across Base, New, and H, with performance improvements of 1.5%, 0.7%, and 0.9%, respectively.

| Datasets | Sets | CoOp (ICCV22) | CoCoOp (CVPR22) | ProGrad (ICCV23) | KgCoOp (ICCV23) | RPO (ICCV23) | PLOT (ICLR23) | LFA (ICCV23) | MaPLe (CVPR23) | DePT (CVPR24) | PromptSRC (ICCV23) | TCP (CVPR24) | DPC (CVPR25) | PAT |
|---|---|---|---|---|---|---|---|---|---|---|---|---|---|---|
| Average | Base | 82.4 | 80.5 | 82.5 | 80.7 | 81.1 | 84.0 | 83.6 | 82.3 | 83.6 | 84.1 | 84.1 | **85.9** | 85.6 |
| | New | 68.0 | 71.7 | 70.8 | 73.6 | 75.0 | 71.7 | 74.6 | 75.1 | 75.0 | 75.0 | 75.4 | 73.3 | **76.1** |
| | H | 74.5 | 75.8 | 76.2 | 77.6 | 77.8 | 77.4 | 78.8 | 78.5 | 79.1 | 79.3 | 79.5 | 79.1 | **80.4** |
| ImageNet | Base | 76.5 | 76.0 | 77.0 | 75.8 | 76.6 | 77.3 | 76.9 | 76.7 | 77.0 | 77.8 | 77.3 | 77.9 | **78.0** |
| | New | 66.3 | 70.4 | 66.7 | 70.0 | **71.6** | 69.9 | 69.4 | 70.5 | 70.1 | 70.7 | 69.9 | 68.0 | 70.5 |
| | H | 71.0 | 73.1 | 71.5 | 72.8 | 74.0 | 73.4 | 72.9 | 73.4 | **74.1** | 73.4 | 73.4 | 72.6 | 74.1 |
| Caltech101 | Base | 97.8 | 98.0 | 98.0 | 97.7 | 98.0 | 98.5 | 98.4 | 97.7 | 98.3 | 98.1 | 98.2 | 98.6 | **98.8** |
| | New | 93.3 | 93.8 | 93.9 | 94.4 | 94.4 | 92.8 | 93.9 | 94.4 | 94.6 | 93.9 | **94.7** | 94.5 | 94.1 |
| | H | 95.5 | 95.8 | 95.9 | 96.0 | 96.0 | 95.6 | 96.1 | 96.0 | 96.4 | 96.0 | **96.5** | 96.5 | 96.4 |
| OxfordPets | Base | 94.5 | 95.2 | 95.1 | 94.7 | 94.6 | 94.5 | 95.1 | 95.4 | 94.3 | 95.5 | 94.7 | 95.8 | **95.8** |
| | New | 96.0 | 97.7 | 97.6 | 97.8 | 97.5 | 96.8 | 96.2 | **97.8** | 97.2 | 97.4 | 97.2 | 97.7 | 97.4 |
| | H | 95.2 | 96.4 | 96.3 | 96.2 | 96.1 | 95.7 | 95.7 | **96.6** | 95.8 | 96.4 | 95.9 | 96.7 | 96.6 |
| Cars | Base | 75.7 | 70.5 | 77.7 | 71.8 | 73.9 | 79.1 | 76.3 | 72.9 | 79.1 | 78.4 | 80.8 | 79.6 | **81.5** |
| | New | 67.5 | 73.6 | 68.6 | 75.0 | **75.5** | 74.8 | 74.9 | 74.0 | **75.5** | 74.7 | 74.1 | 71.2 | 73.5 |
| | H | 71.4 | 72.0 | 72.9 | 73.4 | 74.7 | 76.9 | 75.6 | 73.5 | **77.3** | 75.5 | **77.3** | 75.2 | **77.3** |
| Flowers | Base | 97.3 | 94.9 | 95.5 | 95.0 | 94.1 | 97.9 | 97.3 | 95.9 | 98.0 | 97.9 | 97.7 | 98.2 | **98.2** |
| | New | 67.1 | 71.8 | 71.9 | 74.7 | 76.7 | 73.5 | 75.4 | 72.5 | 76.4 | 76.8 | 75.6 | 72.7 | **77.3** |
| | H | 79.4 | 81.7 | 82.0 | 83.7 | 84.5 | 84.0 | 85.0 | 82.6 | 85.8 | 86.1 | 85.2 | 83.5 | **86.5** |
| Food101 | Base | 89.4 | 90.7 | 90.4 | 90.5 | 90.3 | 89.8 | 90.5 | 90.7 | 90.5 | 90.6 | 90.6 | **91.4** | 90.5 |
| | New | 88.8 | 91.3 | 89.6 | 91.7 | 90.8 | 91.4 | 91.5 | **92.1** | 91.6 | 91.5 | 91.4 | 90.5 | 91.2 |
| | H | 89.1 | 91.0 | 90.0 | 91.1 | 90.6 | 90.6 | 91.0 | **91.4** | 91.1 | 91.1 | 91.0 | 90.9 | 90.8 |
| Aircraft | Base | 39.7 | 33.4 | 40.5 | 36.2 | 37.3 | 42.1 | 41.5 | 37.4 | 43.2 | 42.3 | 42.0 | **49.5** | 46.2 |
| | New | 31.2 | 23.7 | 27.6 | 33.6 | 34.2 | 33.7 | 32.3 | 35.6 | 34.8 | 37.0 | 34.4 | 34.0 | **37.4** |
| | H | 35.0 | 27.7 | 32.8 | 34.8 | 35.7 | 37.5 | 36.3 | 36.5 | 38.6 | 39.5 | 37.8 | 40.4 | **41.3** |
| SUN397 | Base | 80.9 | 79.7 | 81.3 | 80.3 | 80.6 | 82.2 | 82.1 | 80.8 | 82.3 | 82.8 | 82.6 | 82.0 | **82.9** |
| | New | 68.3 | 76.9 | 74.2 | 76.5 | 77.8 | 73.6 | 77.2 | 78.7 | 77.8 | **79.0** | 78.2 | 75.9 | 78.8 |
| | H | 74.1 | 78.3 | 77.6 | 78.4 | 79.2 | 77.7 | 79.6 | 79.8 | 80.0 | **80.9** | 80.4 | 78.9 | 80.8 |
| DTD | Base | 80.0 | 77.0 | 77.4 | 77.6 | 76.7 | 82.0 | 81.3 | 80.4 | 82.2 | 82.6 | 82.8 | **85.5** | 85.3 |
| | New | 48.6 | 56.0 | 52.4 | 55.0 | 62.1 | 43.8 | 60.6 | 59.2 | 59.1 | 57.5 | 58.1 | 55.6 | **63.5** |
| | H | 60.5 | 64.9 | 62.5 | 64.4 | 68.6 | 57.1 | 69.5 | 68.2 | 68.8 | 67.8 | 68.3 | 67.4 | **72.8** |
| EuroSAT | Base | 90.1 | 87.5 | 90.1 | 85.6 | 86.6 | 93.7 | 93.4 | 94.1 | 89.0 | 92.4 | 91.6 | **98.3** | 94.8 |
| | New | 53.0 | 60.0 | 60.9 | 64.3 | 69.0 | 62.7 | 71.2 | 73.2 | 71.1 | 68.4 | **74.7** | 72.2 | 74.4 |
| | H | 66.7 | 71.2 | 72.7 | 73.5 | 76.8 | 75.1 | 80.8 | 82.3 | 79.0 | 78.6 | 82.3 | 83.3 | **83.4** |
| UCF101 | Base | 84.5 | 82.3 | 84.3 | 82.9 | 83.7 | 86.6 | 87.0 | 83.0 | 85.8 | 86.9 | 87.1 | 88.1 | **89.2** |
| | New | 67.4 | 73.5 | 74.9 | 76.7 | 75.4 | 75.9 | 77.5 | 78.7 | 77.2 | 78.3 | **80.8** | 74.2 | 79.3 |
| | H | 75.0 | 77.7 | 79.4 | 79.7 | 79.3 | 80.9 | 82.0 | 80.8 | 81.3 | 82.4 | 83.8 | 80.5 | **84.0** |

## 4.2 BASE-TO-NEW GENERALIZATION

To evaluate the generalization ability of PAT, we equally split each dataset into base and new classes. The model is trained using the base classes in a 16-shot setting and evaluated on new classes. To simultaneously evaluate the fitting ability, generalization capability, and overall performance, we report the classification accuracy for both base classes and new classes, as well as their harmonic mean. Table 2 shows that PAT achieves state-of-the-art performance on 9 out of 11 datasets and is competitive on the remaining SUN397 and Flowers datasets. Compared with CoOp Zhou et al. (2022b), PAT achieves an accuracy gain of 5.9% on average and 3.2% and 8.1% on the base and new classes, respectively. Furthermore, PAT outperforms the state-of-the-art TCP by 0.9% on average (80.4% vs. 79.5%), 1.5% on the base classes (85.6% vs. 84.1%), and 0.7% on the new classes (76.1% vs. 75.4%). These results demonstrate that PAT achieves better fitting capability and generalization ability compared to existing methods.

## 4.3 FEW-SHOT CLASSIFICATION

To better validate the ability of the proposed PAT in transfer learning with limited data, we perform few-shot classification on 11 datasets. All the methods were trained using $K$-shot training images and corresponding class labels, and evaluated on test sets that share the same class space as the training sets. Following previous approaches, we present classification performance on 4-shots. Table 3 shows that PAT achieves the best performance in 8 out of 11 datasets. For example, in DTD, we improved performance from 64% to 65.4%; in EuroSAT, from 77.4% to 85.3%; and in SUN397, from 72.8% to 74.0%. Overall, PAT shows a 1.5% improvement compared to the previous state-of-the-art, providing strong evidence of its capability for downstream transfer learning with limited samples.

Table 3: Accuracy (%) for few-shot classification. PAT achieved state-of-the-art performance in 4-shot settings, delivering an absolute performance improvement of 1.5% compared to TCP.

| Datasets | CLIP | CoOp | CoCoOp | ProGrad | KgCoOp | MaPLe | TIP-Adapter-F | DAPT | PromptSRC | PLOT | TaskRes | TCP | PAT |
|---|---|---|---|---|---|---|---|---|---|---|---|---|---|
| ImageNet | 66.7 | 69.4 | 70.6 | 70.2 | 70.2 | 70.7 | **70.8** | **70.8** | **70.8** | 70.4 | 62.9 | 70.5 | **70.8** |
| Caltech101 | 93.3 | 94.4 | 95.0 | 94.9 | 94.7 | 94.3 | 94.8 | 94.2 | 94.8 | 95.1 | 94.7 | 95.0 | **95.5** |
| OxfordPets | 89.1 | 91.3 | 93.0 | 93.2 | 93.2 | 92.1 | 92.3 | 92.2 | 93.2 | 92.6 | 92.0 | 91.9 | **93.5** |
| StanfordCars | 65.7 | 72.7 | 69.1 | 71.8 | 72.0 | 68.7 | 74.4 | 74.4 | 71.8 | 74.9 | 75.9 | **76.3** | 75.7 |
| Flowers | 70.7 | 91.1 | 82.6 | 90.0 | 90.7 | 80.8 | 93.0 | 92.4 | 91.3 | 92.9 | 91.5 | **94.4** | 93.7 |
| Food101 | 85.9 | 82.6 | 86.6 | 85.8 | 86.6 | **86.9** | 86.2 | 83.6 | 86.1 | 86.5 | 86.0 | 85.3 | 86.3 |
| Aircraft | 24.9 | 33.2 | 30.9 | 32.9 | 32.5 | 29.0 | 35.5 | 32.5 | 32.8 | 35.3 | 33.8 | 36.2 | **38.0** |
| SUN397 | 62.6 | 70.1 | 70.5 | 71.2 | 71.8 | 71.5 | 70.7 | 72.2 | 72.8 | 70.4 | 72.7 | 72.1 | **74.0** |
| DTD | 44.3 | 58.6 | 54.8 | 57.7 | 58.3 | 54.7 | 61.7 | 61.4 | 60.6 | 62.4 | 59.6 | 64.0 | **65.4** |
| EuroSAT | 48.3 | 68.6 | 63.8 | 70.8 | 71.1 | 54.9 | 78.3 | 72.7 | 75.0 | 80.7 | 72.9 | 77.4 | **85.3** |
| UCF101 | 67.6 | 77.4 | 75.0 | 77.8 | 78.4 | 73.7 | 79.7 | 79.4 | 79.4 | 79.8 | 76.1 | 80.8 | **81.7** |
| Average | 65.4 | 73.6 | 72.0 | 74.2 | 74.5 | 70.7 | 76.1 | 75.1 | 75.3 | 76.5 | 74.4 | 76.7 | **78.2** |

Table 4: Ablation study on the hyper-parameters $\alpha$ and $r$ in adapter configuration.

| $\alpha$ | $r$ | EuroSAT | | | DTD | | | UCF101 | | | Flowers | | | Pets | | | Cars | | |
|---|---|---|---|---|---|---|---|---|---|---|---|---|---|---|---|---|---|---|---|
| | | Base | New | H | Base | New | H | Base | New | H | Base | New | H | Base | New | H | Base | New | H |
| 0.1 | 2 | 94.1 | 66.5 | 77.8 | 83.9 | 63.6 | 72.4 | 88.3 | 79.2 | 83.5 | 98.0 | 76.7 | 86.0 | 95.5 | 97.4 | 96.4 | 78.9 | 74.3 | 76.5 |
| 0.1 | 4 | 93.7 | 72.0 | 81.4 | 84.1 | 63.4 | 72.3 | 88.2 | 78.4 | 83.0 | 98.3 | 76.9 | 86.3 | 95.7 | 97.3 | 96.5 | 79.6 | 74.2 | 76.8 |
| 0.1 | 8 | 94.1 | 74.4 | 83.1 | 84.3 | 62.0 | 71.5 | 89.5 | 79.6 | 84.2 | 98.5 | 76.9 | 86.4 | 95.8 | 97.5 | 96.6 | 80.6 | 73.7 | 77.0 |
| 0.1 | 16 | 94.8 | 74.4 | 83.4 | 85.3 | 63.5 | 72.8 | 89.2 | 79.3 | 84.0 | 98.2 | 77.3 | 86.5 | 95.8 | 97.4 | 96.6 | 81.5 | 73.5 | 77.3 |
| 10.0 | 16 | 96.0 | 68.0 | 79.6 | 84.8 | 58.9 | 69.5 | 87.0 | 74.7 | 80.4 | 98.0 | 73.4 | 83.9 | 95.3 | 97.0 | 96.1 | 77.4 | 73.7 | 75.5 |
| 1.0 | 16 | 96.2 | 65.2 | 77.7 | 83.9 | 57.9 | 68.5 | 87.4 | 78.1 | 82.5 | 98.2 | 73.5 | 84.0 | 95.8 | 96.8 | 96.3 | 77.4 | 74.1 | 75.7 |
| 0.1 | 16 | 94.8 | 74.4 | 83.4 | 85.3 | 63.5 | 72.8 | 89.2 | 79.3 | 84.0 | 98.2 | 77.3 | 86.5 | 95.8 | 97.4 | 96.6 | 81.5 | 73.5 | 77.3 |
| 0.01 | 16 | 92.8 | 75.9 | 83.5 | 83.7 | 60.3 | 70.0 | 86.4 | 78.2 | 82.1 | 98.4 | 76.6 | 86.1 | 96.0 | 97.2 | 96.6 | 77.5 | 75.6 | 76.5 |

## 4.4 Cross-Dataset Generalization

Cross-dataset generalization is evaluated to further validate the generalization ability of PAT, considering that the base and new classes sampled from the same datasets are similar in data distribution in base-to-new generalization. Unlike previous studies, we aim to verify whether models maintain strong generalization abilities after downstream transfer under truly small-scale data with limited samples. The full results please refer to Appendix.

## 4.5 Ablation Studies

Ablation studies are performed on base-to-new generalization on EuroSAT, DTD, UCF101, Flowers, Pets, and Cars to validate the loss function, adapter configuration, and prompt length. We evaluate on each dataset using three random seeds, and report average accuracy for base classes, new classes, and their harmonic mean. The results of ablation on adapter configuration and prompt length please refer to appendix.

## 4.6 Adapter Configuration

Since PAT relies on adapters to constrain the update of prompt learning, we perform ablation experiments on the scaling factor $\alpha$ and hidden dimensions $r$ of the adapter. Table 4 shows that the configuration with $\alpha = 0.1$ and $r = 16$ achieves the best comprehensive performance overall. However, other hyperparameter combinations can outperform this configuration on specific datasets. For instance, $\alpha = 0.1$ and $r = 8$ perform better on UCF101, while $\alpha = 0.01$ and $r = 16$ achieve superior results on EuroSAT. Note that, compared to the hidden dimension $r$, the scaling factor $\alpha$ has a more significant impact on performance across all three datasets.

## 5 Conclusion

In this paper, we propose a novel prompt learning approach based on pre-adjustment, post-adjustment, and contrastive learning. To further enhance the fitting ability and generalization of current prompt learning methods, we employ adapter-based feature adaptation as a post-adjustment to integrate the strengths of Prompts and Adapters, thereby achieving a better balance between fitting capability and generalization ability. Furthermore, we utilize a tolerance regularization to bring known samples closer to text representations while penalizing noise samples against existing text representations, resulting in a more robust multimodal classifier. Our extensive experimental results, including Base-to-New, Cross-dataset, and few-shot evaluations, demonstrate that our proposed method, PAT, achieves significant advancements in both fitting performance and generalizability compared to previous SOTA.

## 6 ETHICS STATEMENT

This work adheres to the ICLR Code of Ethics. All datasets used such as ImageNet, Caltech, Food , were sourced in compliance with relevant usage guidelines, ensuring no violation of privacy. We have taken care to avoid any biased or discriminatory outcomes in our research process. No personally identifiable information was used. We arecommited to maintaining transparency and integrity throughout the research process.

## 7 REPRODUCIBILITY STATEMENT

We have made every effort to ensure that the results presented in this paper are reproducible. The experimental setup, including training steps, model configurations, and hardware details is described in detail. Additionally, all datasets used such as ImageNet, Caltech, Food, are publicly available, ensuring consistent and reproducible evalutaion results.

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

# A APPENDIX

## A.1 THE USAGE OF LARGE LANGUAGE MODELS

During the preparation of this manuscript, Large Language Models was utilized primarily for polishing and enhancing the readability in various sections. Specifically, LLMs assisted me in checking grammar errors, improving phrasing, and enhancing the overall fluency of the writing. And it should be emphasized that LLMs did not contribute to the development of methodologies or the design of experiments; all research ideas, data analyses, and experiments were carried out by the authors of the paper.

## A.2 PROOF OF PROPOSITION 3.1

When we only consider the positive samples Then we have the sigmoid loss function as

$$\mathcal{L} = -\frac{1}{|B|} \sum_{i=1}^{|B|} \log \kappa(cf_i t_i - b). \tag{17}$$

where $\kappa$ is the sigmoid function as $\kappa(u) = \frac{1}{1+e^{-u}}$ Besids, the cross entropy can be simplified as

$$H(\hat{y}, y) = -\frac{1}{|B|} \sum_{i=1}^{|B|} \log \frac{e^{cf_i t_{k^*}+b}}{\sum_{j=1}^{K} e^{cf_i t_j+b}}. \tag{18}$$

Considering that only on positive textual label for each image sample. Besides, Similarity scores for all negative samples are constant as $cf_i t_j + b = 0 (j \neq k^*)$. Then we have

$$\sum_{j=1}^{K} e^{cf_i t_j+b} = e^{cf_i t_{k^*}+b} + (K-1)e^0 = e^{cf_i t_{k^*}+b} + (K-1). \tag{19}$$

Then the cross entropy degrads into:

$$H(\hat{y}, y) = -\frac{1}{|B|} \sum_{i=1}^{|B|} \log \frac{e^{cf_i t_{k^*}+b}}{e^{cf_i t_{k^*}+b} + (K-1)}. \tag{20}$$

When we further consider the binary classification, i.e., whether the image feature is aligned with the textual description, we have

$$H(\hat{y}, y) = -\frac{1}{|B|} \sum_{i=1}^{|B|} \log \frac{e^{cf_i t_{k^*}+b}}{e^{cf_i t_{k^*}+b} + 1} = -\frac{1}{|B|} \sum_{i=1}^{|B|} \log \kappa(cf_i t_{k^*} + b). \tag{21}$$

## A.3 DOMAIN GENERALIZATION

Domain generalization experiments involve training a model on the source domain and testing it on the target domain, making them useful for evaluating model generalization. Therefore, we train PAT under the ImageNet 16-shot setting and test it on ImageNetV2, ImageNet-Sketch, ImageNet-A, and ImageNet-R. The final results are reported as the average performance across these five datasets. As shown in Table 5, PAT still holds state-of-the-art performance.

## A.4 CROSS-DATASET GENERALIZATION

In this experiment, all methods are trained on the base classes of the DTD and EuroSAT datasets and all classes of ImageNet in 16-shot settings under three distinct random seeds, and subsequently evaluated on all categories of the other datasets. We compare the proposed PAT with Zero-shot CLIP and TCP that demonstrates robust performance in base-to-new generalization.

Table 6 shows that, when trained on these cross-distribution few-shot datasets, TCP is inferior to zero-shot CLIP in most cases. Notably, compared with zero-shot CLIP, TCP suffers accuracy loss of 7.2% on EuroSAT and 2.0% on DTD. In contrast, PAT surpasses zero-shot CLIP and outperforms TCP by 7.7% and 3.3% on these two datasets, respectively. This further demonstrates the superior generalization ability of PAT.

| Datasets | ImageNet | -V2 | -S | -A | -R | Avg. |
|---|---|---|---|---|---|---|
| CoCoOp | 71.0 | 64.1 | 48.8 | 50.6 | 76.2 | 62.1 |
| ProGrad | 72.2 | 64.7 | 47.6 | 49.4 | 74.6 | 61.7 |
| KgCoOp | 71.2 | 64.1 | 49.0 | 50.7 | 76.7 | 62.3 |
| MaPLe | 70.7 | 64.1 | 49.2 | 50.9 | 77.0 | 62.4 |
| DAPT | 71.7 | 64.5 | 49.5 | 51.1 | 76.3 | 62.6 |
| TCP | 71.2 | 64.6 | 49.5 | **51.2** | 76.7 | 62.6 |
| PromptSRC | 71.3 | 64.4 | **49.6** | 50.9 | **77.8** | 62.8 |
| PAT | **72.8** | **66.5** | 49.4 | 49.0 | 77.1 | **63.0** |

Table 5: Performance comparison across different methods on Domain Generalization Experiment. PAT achieved state-of-the-art performance, delivering an absolute performance improvement of 0.2% compared to PromptSRC.

Table 6: Accuracy (%) for cross-dataset generalization. PAT achieved state-of-the-art performance in all settings.

| Parameter-Efficient Fine-Tuning On DTD Base Classes | | | | | | | | | | |
|---|---|---|---|---|---|---|---|---|---|---|
| Methods | Caltech101 | OxfordPets | Cars | Flowers | Food101 | Aircraft | SUN397 | EuroSAT | UCF101 | ImageNet | Average |
| CLIP | 93.3 | 89.1 | 65.6 | 70.7 | 85.9 | 24.7 | 62.6 | 48.3 | 67.6 | 72.4 | 68.0 |
| TCP | 91.6 | 86.8 | 64.7 | 68.5 | 85.2 | 20.5 | 62.1 | 46.4 | 68.2 | 65.6 | 66.0 |
| PAT | 96.7 | 89.4 | 60.3 | 66.3 | 87.9 | 22.4 | 72.0 | 57.5 | 68.6 | 71.6 | **69.3** |

| Parameter-Efficient Fine-Tuning On EuroSAT Base Classes | | | | | | | | | | |
|---|---|---|---|---|---|---|---|---|---|---|
| Methods | Caltech101 | OxfordPets | Cars | Flowers | Food101 | Aircraft | SUN397 | DTD | UCF101 | ImageNet | Average |
| CLIP | 93.3 | 89.1 | 65.6 | 70.7 | 85.9 | 24.7 | 62.6 | 44.1 | 67.6 | 72.4 | 67.6 |
| TCP | 86.4 | 82.8 | 61.4 | 65.1 | 83.3 | 16.5 | 51.6 | 34.8 | 63.3 | 58.8 | 60.4 |
| PAT | 96.8 | 87.0 | 60.4 | 63.3 | 88.9 | 21.4 | 70.7 | 53.9 | 68.9 | 69.2 | **68.1** |

| Parameter-Efficient Fine-Tuning On ImageNet all Classes | | | | | | | | | | |
|---|---|---|---|---|---|---|---|---|---|---|
| Methods | Caltech101 | OxfordPets | Cars | Flowers | Food101 | Aircraft | SUN397 | DTD | EuroSAT | UCF101 | Average |
| CLIP | 93.3 | 89.1 | 65.6 | 70.7 | 85.9 | 24.7 | 62.6 | 44.1 | 48.3 | 67.6 | 65.2 |
| CoOp | 93.7 | 89.1 | 64.5 | 68.7 | 85.3 | 18.5 | 64.2 | 41.9 | 46.4 | 66.6 | 63.9 |
| ProGrad | 91.5 | 89.6 | 62.4 | 67.9 | 85.4 | 20.2 | 62.5 | 39.4 | 43.5 | 64.3 | 62.7 |
| KgCoOp | 93.9 | 89.8 | 65.4 | 70.0 | 86.4 | 22.5 | 66.2 | 46.4 | 46.0 | 68.5 | 65.5 |
| DePT | 94.2 | 90.0 | 65.6 | 70.6 | 86.4 | 23.3 | 66.7 | 46.0 | 43.5 | 69.3 | 65.6 |
| VPT | 93.7 | 89.3 | 65.5 | 70.2 | 86.3 | 22.1 | 66.6 | 46.9 | 47.4 | 67.2 | 65.5 |
| PLOT | 92.1 | 90.1 | 65.7 | 69.2 | 86.2 | **25.0** | 61.7 | 38.6 | 47.8 | 67.0 | 64.3 |
| PromptSRC | 93.6 | 90.3 | 65.7 | 70.3 | 86.2 | 23.9 | 67.1 | **46.9** | 45.5 | 68.8 | 65.8 |
| MaPLe | 93.5 | 90.5 | 65.6 | **72.2** | 86.2 | 24.7 | 67.0 | 46.5 | 48.1 | 68.7 | 66.3 |
| DAPT | 93.5 | 90.7 | **65.9** | 71.7 | 86.1 | 23.0 | 67.0 | 44.0 | **52.5** | 68.7 | 66.3 |
| TCP | **94.0** | **91.3** | 64.7 | 71.2 | **86.7** | 23.5 | 67.2 | 44.4 | 51.5 | 68.7 | 66.3 |
| PAT | 93.4 | 90.2 | 65.8 | 71.3 | 86.0 | 24.5 | **67.6** | 46.1 | 50.8 | **68.9** | **66.5** |

## A.5 ABLATION ON ADAPTER CONFIGURATION

Since PAT relies on adapters to constrain the update of prompt learning, we perform ablation experiments on the scaling factor $\alpha$ and hidden dimensions $r$ of the adapter. Table 4 shows that the configuration with $\alpha = 0.1$ and $r = 16$ achieves the best comprehensive performance overall. However, other hyperparameter combinations can outperform this configuration on specific datasets. For instance, $\alpha = 0.1$ and $r = 8$ perform better on UCF101, while $\alpha = 0.01$ and $r = 16$ achieve superior results on EuroSAT. Note that, compared to the hidden dimension $r$, the scaling factor $\alpha$ has a more significant impact on performance across all three datasets.

### A.5.1 ABLATION ON LOSS FUNCTION

We first validate the effectiveness of the proposed loss function, including the alignment loss to constrain pre-adjustment and post-adjustment and the tolerance regularization for constructing a robuster classifier. Table 7 shows that the absence of the tolerance regularization results in a 1.4% decline in overall performance on EuroSAT, with accuracy decreasing by 1.3% on the Base category and 1.4% on the New category. On DTD, the overall performance drops by 0.4%, with a 1.3% decrease in the Base category. Similarly, when the PP loss is removed, the accuracy on EuroSAT decreases by 2.7% in the New category and 1.6% overall. On DTD, the Base, New, and overall performance decrease by 0.3%, 2.3%, and 1.6%, respectively. In this setting, the baseline removes both the PP and Tol losses. Compared to the baseline, both PP and Tol bring significant improvements. Specifically, the PP loss improves overall performance by 8.3%, 6.0%, and 1.7%

Table 7: Ablation study on the loss function. B, PP, and Tol stand for Baseline, Pre-Post, and Tolerance, respectively.

| B | PP | Tol | EuroSAT | | | DTD | | | UCF101 | | |
|---|---|---|---|---|---|---|---|---|---|---|---|
| | | | Base | New | H | Base | New | H | Base | New | H |
| ✓ | | | 96.6 | 60.4 | 74.3 | 84.7 | 54.6 | 66.4 | 88.5 | 76.5 | 82.1 |
| ✓ | ✓ | | 93.5 | 73.0 | 82.0 | 84.0 | 63.6 | 72.4 | 89.0 | 79.2 | 83.8 |
| ✓ | | ✓ | 95.2 | 71.7 | 81.8 | 85.0 | 61.2 | 71.2 | 89.0 | 80.5 | 84.5 |
| ✓ | ✓ | ✓ | 94.8 | 74.4 | 83.4 | 85.3 | 63.5 | 72.8 | 89.2 | 79.3 | 84.0 |

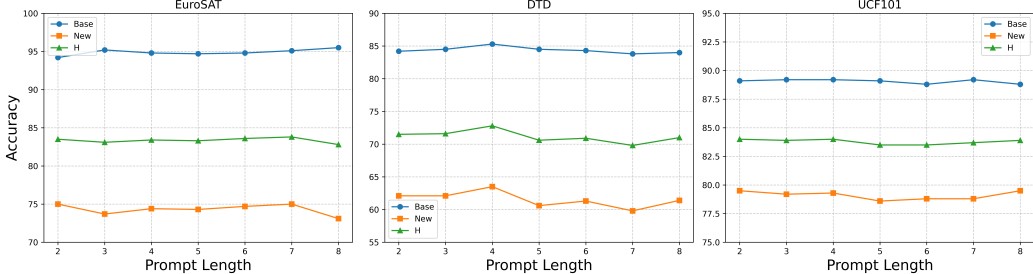

Figure 4: Ablation study on prompt length ranging from 2 to 8.

on EuroSAT, DTD, and UCF101, respectively, while the Tol loss leads to gains of 8.1%, 4.8%, and 2.4% on the these datasets. These results demonstrate the effectiveness of our method and further validate that adjusting the alignment before integration is superior to directly combining Adapter and Prompt approaches.

## A.6 ABLATION ON PROMPT LENGTH

We investigate the impact of prompt length under the Base-to-New configuration. We compare the performance effects of prompt lengths ranging from 2 to 8. Figure 4 shows that PAT is generally insensitive to the choice of prompt length. In previous experiments, we fixed the prompt length to 4. However, when the prompt length is set to 2, 6, or 7, PAT's performance on EuroSAT can be further improved, and a length of 2 achieves better performance on UCF101.

