# OpenReview forum: "Bootstrap Prompt Learning with Feature Adaptation for Vision-Language Efficient Tuning"
_ICLR.cc/2026/Conference — ICLR 2026 Conference Withdrawn Submission_

### Official Review · Reviewer_WXzj · 2025-10-27

**Soundness:** 1
**Presentation:** 1
**Contribution:** 1
**Rating:** 2
**Confidence:** 5

**Summary:**

This paper proposes **PAT (adaPter bootstrApped prompt contrastive Tuning)**, a parameter-efficient fine-tuning framework that aims to combine the strengths of **Prompt Learning** (good at generalization) and **Adapter Tuning** (good with fitting).
The method introduces two main components:
(1) **Pre-Post Alignment**, which aligns prompt-based and adapter-based representations through an MSE loss;
and (2) **Tolerance Regularization**, a sigmoid-based binary contrastive loss intended to reduce overfitting from cross-entropy.
Experiments on multiple vision-language benchmarks show some gains (around 1%) over prior PEFT baselines.

While the paper is easy to understand, its main issue lies in the limited novelty. In addition, there are several methodological errors in the descriptions and assumptions within the methodology section, and the experimental improvements are very marginal.

**Strengths:**

The paper attempt to combine prompt learning and adapter tuning, aiming to balance generalization and fitting within a unified PEFT framework.

**Weaknesses:**

1**Motivation is not clear**
The motivation is not convincing. Why does prompt learning exhibit strong generalization ability, while adapter-based methods have strong fitting ability? Any other quantitative results or related literature to show that?

2 **Novelty is Limited**
The work seems to be a direct combination of prompt tuning and adapter tuning. Even when combined, the model complexity and training cost are higher than using either alone. What's the additional benefits beyond performance gains on this combination?

3 **Some Assumptions are Not True**
In Section 3.2.1 (*Pre-Post Alignment*), the authors treat the Adapter and Prompt outputs as probabilistic models and assume \(p(z|x)\) follows a Gaussian distribution — which is unrealistic, since a proper probabilistic model should satisfy the **simplex constraint** (e.g., Dirichlet distribution).

4 **Lack of Justification for “Tolerance Regularization”**
The proposed *Sigmoid-based binary contrastive loss* is essentially the standard BCE/logistic contrastive loss already used in CLIP. The claimed theorem is trivial, and there is no clear explanation of what “tolerance” means or how it improves generalization. The use of a −0.01 scaling factor is unjustified.

**Questions:**

Why can the proposed tolerance regularization improve generalization performance? Please explain the underlying principle in detail. In addition, the authors mention in Line 310 that it is non-differentiable — how, then, is this loss optimized during gradient descent?

---

### Official Review · Reviewer_pco1 · 2025-10-29

**Soundness:** 2
**Presentation:** 2
**Contribution:** 2
**Rating:** 2
**Confidence:** 4

**Summary:**

This paper proposes a framework named adaPter bootstrApp prompt contrastive Tuning (PAT) for efficient fine-tuning of vision-language models (VLMs) such as CLIP. The method combines prompt learning (as pre-adjustment) with adapter-based tuning (as post-adjustment), aiming to balance generalization and fitting capacity. Additionally, the authors introduce a tolerance regularization mechanism to mitigate overfitting caused by exclusive cross-entropy loss. Experiments across multiple benchmarks demonstrate improved performance over existing PEFT methods.

**Strengths:**

The paper is  clearly structured, with comprehensive experimental evaluations across 11 datasets.

The integration of prompt learning and adapter tuning is systematically implemented and supported by empirical results.

The regularization is a thoughtful addition to address overfitting and improve generalization.

**Weaknesses:**

1. The core techniques prompt learning, adapter tuning, and contrastive regularization, are all based on existing methods. The paper primarily repurposes these techniques for VLM adaptation without introducing fundamentally novel algorithms or theoretical insights.

2. The proposed pre-post alignment is essentially a feature-level MSE loss between prompt and adapter outputs, which is a straightforward application of existing alignment strategies. e.g., PACE: Marrying generalization in PArameter-efficient fine-tuning with Consistency rEgularization. (NeurIP 2024)

3. The tolerance regularization, while useful, closely resembles binary contrastive loss and does not offer a significant conceptual leap.

4. The paper lacks a deeper analysis or ablation to justify why the combination of prompt and adapter tuning leads to synergistic gains beyond additive effects.

**Questions:**

Can you clarify what distinguishes your pre-post alignment from prior works that align multimodal features using MSE or KL divergence?

How does your tolerance regularization differ from existing contrastive or binary cross-entropy losses in terms of optimization behavior or theoretical properties?

---

### Official Review · Reviewer_36dx · 2025-10-31

**Soundness:** 3
**Presentation:** 3
**Contribution:** 1
**Rating:** 2
**Confidence:** 3

**Summary:**

The manuscript proposes PAT for efficient adaptation of CLIP-style vision–language models (VLMs). PAT (i) combines prompt learning ("pre-adjustment") with adapters ("post-adjustment"), aligned via a KL/MSE objective, and (ii) adds a "tolerance regularization" term, formulated as a pairwise sigmoid (binary) contrastive loss intended to mitigate softmax bias and improve base-to-novel generalization. Experiments on 11 datasets report small but consistent gains in base-to-new and few-shot settings, using ViT-B/16, with results averaged over three runs.

**Strengths:**

1. Treating prompts as pre-adjustment and adapters as post-adjustment is a clear, implementation-friendly decomposition, with minimal surgery to a frozen CLIP.
2. The pre–post alignment reduces to MSE under Gaussian assumptions, which is easy to reproduce and ablate.
3. Base-to-new and 4-shot results are reported with 3 seeds and show modest average improvements.

**Weaknesses:**

1. Tolerance loss overlaps SigLIP. The pairwise sigmoid contrastive objective is very similar to SigLIP; what is substantively new beyond constants or the "noise" sampling? Please provide a SigLIP drop-in ablation. If PAT still wins, analyze why (e.g., interaction with pre/post alignment).
2. Reducing KL to MSE assumes equal variances/linearity, but the paper does not test whether feature-space alignment is preferable to logit/probability alignment. Please compare feature vs. logit vs. prob alignment; add a small sensitivity study to the alignment weight and to variance/temperature settings.
3. Current ablations do not isolate capacity vs. loss vs. alignment. Please conduct more ablations about prompt length, adapter width, alignment weight, and tolerance weight across multiple datasets, with error bars. Include a minimal effective configuration.
4. Reference. The paper claims "the first successful attempt to simultaneously exploit the advantages of prompt learning and adapter tuning," yet there exist unified or hybrid frameworks that combine adapters with prompt-style conditioning and foundation-model priors (e.g., APoLLo[1]: unified adapter + LLM/diffusion-based prompt augmentation; also CLIP-Adapter as an earlier, strong adapter baseline). APoLLo explicitly targets the same CLIP-transfer regime with adapters and prompt augmentation. Please discuss PAT’s novelty relative to those designs and, ideally, add them to comparisons.

[1] APoLLo : Unified Adapter and Prompt Learning for Vision Language Models. EMNLP 2023.

**Questions:**

Please see the weakness part.

---

### Note · Authors · 2025-11-13

I have read and agree with the venue's withdrawal policy on behalf of myself and my co-authors.